# Siliceous zeolite-derived topology of amorphous silica

Hirokazu Masai [1✉], Shinji Kohara [2✉], Toru Wakihara[3], Yuki Shibazaki[4], Yohei Onodera [5,15], Atsunobu Masuno [6], Sohei Sukenaga[7], Koji Ohara [8,16], Yuki Sakai [9,10], Julien Haines [11], Claire Levelut[12], Philippe Hébert[13], Aude Isambert [13,17], David A. Keen [14] & Masaki Azuma [9,10]

The topology of amorphous materials can be affected by mechanical forces during compression or milling, which can induce material densification. Here, we show that densified amorphous silica ($SiO_2$) fabricated by cold compression of siliceous zeolite (SZ) is permanently densified, unlike densified glassy $SiO_2$ (GS) fabricated by cold compression although the X-ray diffraction data and density of the former are identical to those of the latter. Moreover, the topology of the densified amorphous $SiO_2$ fabricated from SZ retains that of crystalline SZ, whereas the densified GS relaxes to pristine GS after thermal annealing. These results indicate that it is possible to design new functional amorphous materials by tuning the topology of the initial zeolitic crystalline phases.

[1] Department of Materials and Chemistry, National Institute of Advanced Industrial Science and Technology, 1-8-31 Midorigaoka, Ikeda, Osaka 563-8577, Japan. [2] Center for Basic Research on Materials, National Institute for Materials Science, 1-2-1 Sengen, Tsukuba, Ibaraki 305-0047, Japan. [3] Institute of Engineering Innovation, The University of Tokyo, Yayoi 2-11-16, Bunkyo-ku, Tokyo 113-8656, Japan. [4] Photon Factory, Institute of Materials Structure Science, High Energy Accelerator Research Organization (KEK), 1-1 Oho, Tsukuba, Ibaraki 305-0801, Japan. [5] Institute for Integrated Radiation and Nuclear Science, Kyoto University, 2-1010 Asashiro-nishi, Kumatori-cho, Sennan-gun, Osaka 590-0494, Japan. [6] Graduate School of Engineering, Kyoto University, Kyotodaigaku-katsura, Nishikyo-ku, Kyoto 615-8520, Japan. [7] Institute of Multidisciplinary Research for Advanced Materials, Tohoku University, 2-1-1 Katahira, Aoba-ku, Sendai, Miyagi 980-8577, Japan. [8] Japan Synchrotron Radiation Research Institute (JASRI/SPring-8), Kouto, Sayo-cho, Hyogo 679-5198, Japan. [9] Kanagawa Institute of Industrial Science and Technology (KISTEC), 705-1 Shimoimaizumi, Ebina, Kanagawa 243-0435, Japan. [10] Laboratory for Materials and Structures, Tokyo Institute of Technology, 4259 Nagatsuta, Yokohama, Kanagawa 226-8503, Japan. [11] Institut Charles Gerhardt Montpellier, CNRS, Université de Montpellier, ENSCM, 34293 Cedex 5 Montpellier, France. [12] Laboratoire Charles Coulomb, CNRS, Université de Montpellier, 34095 Montpellier, France. [13] CEA, DAM Le Ripault, F-37260 Monts, France. [14] ISIS Facility, Rutherford Appleton Laboratory, Harwell Campus, Didcot, Oxfordshire OX11 0QX, UK. [15] Present address: Center for Basic Research on Materials, National Institute for Materials Science, 1-2-1 Sengen, Tsukuba, Ibaraki 305-0047, Japan. [16] Present address: Faculty Materials for Energy, Shimane University, 1060 Nishikawatsu-cho, Matsue, Shimane 690-8504, Japan. [17] Present address: Institut de Physique du Globe de Paris (IPGP), Université Paris Cité, Paris, France. ✉email: hirokazu.masai@aist.go.jp; KOHARA.Shinji@nims.go.jp

The properties of solid-state materials are significantly affected by their preparation conditions and chemical compositions. Polymorphisms in crystalline materials with the same chemical composition have been investigated using various approaches[1–4]. In contrast, in non-equilibrium materials, such as glasses, in which various metastable structures exist, structural relaxation by external stimuli is one of the most interesting topics from both scientific and industrial perspectives. This metastability is one of the challenging factors for the structural analysis of glass[5].

Non-equilibrium oxide materials, such as glasses and zeolites, possess nanosized cavities that are specific to their functions. In materials with such large cavities, thermodynamically metastable structures can be formed semipermanently or transiently by applying a much higher pressure than the ambient pressure while simultaneously heating[6–20]. Densified samples fabricated at high pressures exhibit completely different functions from those fabricated under ambient pressure. Greaves et al. predicted that these microporous materials could approach the "perfect" glass compressed sufficiently slowly[6]. Densified glassy silica (GS) is tentatively proposed as an example of a high-pressure-induced densified material because pristine GS possesses large cavities surrounded by –Si–O–Si– rings of varying sizes. Recently, experimental and mathematical approaches have been combined to investigate the behaviours of rings and cavities in amorphous materials[19–22]. Owing to their varied rings and cavities, oxide materials containing many oxygen atoms with lone-pair electrons are interesting materials for study.

From a material densification perspective, zeolites with their open-structured micropores are also interesting targets for controllable cavities[1,2,23–30]. Approximately 260 different zeolite structures are known, ranging from those with one-dimensional channels to those with three-dimensional pores, a number of which are smaller than 1 nm. Zeolites provide another route for preparing distinct amorphous materials via pressure-induced amorphization. Haines et al. reported the densification of amorphous $SiO_2$ by pressurising a single crystal of siliceous MFI zeolite (SZ) to 20 GPa at room temperature (RT) (i.e. cold compression[17–19,31]). The Bragg peaks from the SZ disappeared and broad peaks corresponding to amorphous were observed. However, Onodera et al. reported that the density of densified GS prepared by cold compression decreased over time, i.e. glass prepared by cold compression was not permanently densified[19]. Considering that they used GS as the starting material, it is unclear whether permanent densification could be achieved in the amorphous $SiO_2$ prepared from SZ. We also investigated whether different topologies of SZ could be obtained via ball milling. Mechanical milling is sometimes used to prepare reactive ceramic powders, such as oxides, sulfides, and chalcogenides[32,33]. The ball-milling process is expected to break the cages in the SZ, producing more reactive fragments. This study analysed amorphous $SiO_2$ and SZ to clarify the relationship between the starting materials and the glass structures.

## Results and discussion

### Preservation of cage structure in SZ-derived amorphous $SiO_2$.

Figure 1a shows the X-ray powder diffraction pattern of amorphous $SiO_2$ from SZ, prepared by applying 20 GPa and 7.7 GPa at RT, along with previous data reported for densified GS[19] and the densified amorphous SZ prepared from SZ single crystals[31]. All data, except for those of the reference materials (pristine GS and pristine SZ), were acquired from the samples recovered after densification. The structure factor $S(k)$ of various amorphous materials differs depending on the preparation conditions. This study focused on the first sharp diffraction peak (FSDP), which is referred to as $k_1$ and observed at $k \sim 1.53$ Å$^{-1}$ in the diffraction pattern of pristine GS (Fig. 1b). The FSDP, a signature of intermediate-range ordering in glass, shifts to a higher-$k$ value upon applying pressure, suggesting that the intermediate correlation distances decrease with the reduction in cavity volume. In addition, a pre-peak is observed at $k \sim 0.63$ Å$^{-1}$ in all samples obtained from SZs, and the peak height decreases with increasing pressure. This peak can be referred to as $k_0$ because the FSDP at a higher-$k$ value is typically referred to as $k_1$ and the second principal peak is called $k_2$[34]. The $k_2$ peak is only visible in the neutron diffraction data (Fig. S1) because $k_2$ reflects the packing of oxygen atoms, and relative to silicon, oxygen scatters neutrons better than it scatters X-rays[35]. Notably, the $k_0$ peak was not observed for the GS or the densified GS. Table 1 summarises the starting materials, fabrication conditions, densities, and coherence lengths estimated from diffraction peaks. The density $\rho$ was measured using a He pycnometer or by the analysis of the slope of reduced pair distribution functions $G(r)$ using the equation $\rho = \frac{1}{4\pi} \frac{\partial G(r)}{\partial r}$, where $r$ is a length.

These results clearly indicate that the structures of the densified GS and amorphous $SiO_2$ prepared from the SZ are different. The $k$ value of the $k_0$ peak has similar $k$ value as that observed for the strong Bragg peaks in the X-ray diffraction pattern of the pristine SZ, implying that the topology of the crystalline starting material can be preserved in the amorphous material even after high-pressure treatment. An illustration of the SZ, highlighted by the (101) and (020) planes, is presented in Fig. S2. Notably, the height of $k_0$ increased with $k_1$. Therefore, it is expected that amorphous $SiO_2$ with different topologies can be obtained by selecting appropriate starting materials.

### Permanent densification and relaxation of SZ-derived amorphous $SiO_2$.

Because compression was performed at ambient temperature, it is expected that the long-term thermal stability is also affected by the topology of the samples. To analyse the thermal stability after densification, i.e. the permanency of densification, we measured the diffraction data from the same samples after long delays. Figure 2a, b compares the $S(k)$ values of amorphous $SiO_2$ prepared by cold compression after 11 and 2 years, respectively. Note that the former was obtained from bulk SZ (bSZ) single crystals with typical maximum linear dimensions of 25–80 μm and the latter from a SZ powder (pSZ) with micro-sized grains. Although the two sets of data are similar, the structures of the samples depend on the starting material. Figure 2c, d shows an enlarged portion of $S(k)$ from amorphous $SiO_2$ and the differential $S(k)$ and $\Delta S(k)$ values of bSZ and pSZ. $\Delta S(k)$ is the difference in $S(k)$ between the as-prepared sample and the same sample after an extended period. The positions of the FSDP and the peak at $k_3$ are indicated by the dashed lines in Fig. 2c, d. The amorphous $SiO_2$ from bSZ was very stable and showed no remarkable difference in the diffraction pattern after 11 years (i.e. there was no peak shift in Fig. 2c). In contrast, shifts in FSDP and $k_3$ were observed for pSZ even after 2 years (Fig. 2d). The $k$ value of FSDP decreased, whereas that of $k_3$ increased. This behaviour is comparable to that of $S(k)$ in pristine GS and densified amorphous $SiO_2$ as shown in Fig. 1. It is clear that $\Delta S(k)$, as shown in Fig. 2d, corresponds to the structural relaxation of the densified amorphous $SiO_2$ to pristine GS. Hence, we can conclude that bSZ-derived amorphous $SiO_2$ was permanently densified, whereas the amorphous $SiO_2$ from pSZ was not. Considering that GS densified from bulk GS exhibits permanent densification behaviour with thermal treatment[19], we assume that single crystals are an important starting point for sustaining permanent densification via cold compression. Monolithic materials are expected to be advantageous for efficient densification because they do not require energy to remove grain boundaries or defects.

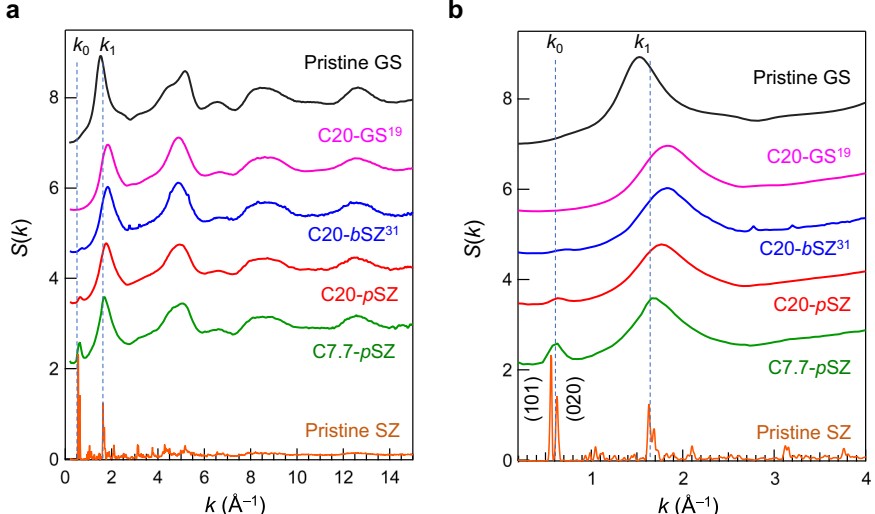

**Fig. 1 Comparison of X-ray diffraction data. a** Total structure factors, $S(k)$, of amorphous $SiO_2$ materials prepared by cold compressions: pristine glassy $SiO_2$ (GS), densified GS after cold compression with 20 GPa (C20-GS), densified amorphous $SiO_2$ from bulk crystal siliceous zeolite after cold compression with 20 GPa (C20-$b$SZ), densified amorphous $SiO_2$ obtained from siliceous zeolite powder by 7.7 GPa and 20 GPa cold compression (C7.7-$p$SZ and C20-$p$SZ, respectively); the dashed lines indicate the position of the scattering vector for $k_0$ and $k_1$ in GS without densification. **b** Enlarged $S(k)$ in the FSDP region of amorphous $SiO_2$.

**Table 1 Structural parameters of amorphous $SiO_2$ (all densified samples are synthesised by cold compression).**

| ID | Starting material | Press condition | Density (g cm$^{-3}$) | $k_0$ (Å$^{-1}$) | $k_1$ (Å$^{-1}$) | $2\pi/k_0$ (Å) | $2\pi/k_1$ (Å) |
|---|---|---|---|---|---|---|---|
| Pristine GS | Glassy $SiO_2$ | NA | 2.2 | — | 1.53 | — | 4.12 |
| C20-GS | Glassy $SiO_2$[19] | 20 GPa | 2.7 | — | 1.85 | — | 3.40 |
| C20-$b$SZ | Bulk siliceous zeolite[31] | 20 GPa | 2.7 | 0.7 | 1.83 | 8.97 | 3.44 |
| C20-$p$SZ | Siliceous zeolite powder | 20 GPa | 2.3 | 0.63 | 1.77 | 9.94 | 3.56 |
| C7.7-$p$SZ | Siliceous zeolite powder | 7.7 GPa | 2.1 | 0.63 | 1.66 | 9.94 | 3.79 |

The mechanism of permanent densification has recently been discussed in terms of the topology[19]. It has been suggested that both ring size distribution and cavity volume are correlated with densification[19]. The ring-size distributions for a series of SZs and GS, calculated based on King's criterion[36], are shown in Fig. 3. The GS data show a variation in the ring size that is topologically disordered[21,22]; however, the SZ data have a large fraction of five-fold rings, which is not representative of topological disorders. Notably, the ring size distributions of both SZ and GS subjected to cold compression at 20 GPa were similar to those of their respective compounds at ambient pressure. The GS results seem to conflict with previous results for densified $SiO_2$ glass[18]. However, although the definition of an $n$-membered ring and the simulation method used in the present study are different from the previous study, we assume that it is difficult to conclude that a remarkable difference in the ring size distribution is observed by the cold compression of the GS.

We also visualised the cavities (highlighted in green) in the SZ and GS, as shown in Fig. 3. The cavity volume ratio (CVR) of pristine GS was 33 vol% at 0 GPa[22]. The CVR was highest for pristine SZ at 0 GPa and lowest for amorphous $SiO_2$ prepared by cold compression at 20 GPa. Furthermore, when comparing the samples prepared by cold compression at 20 GPa, the CVR of the GS is larger than that of amorphous $SiO_2$ from SZ by 3.8%. We suggest that the small fraction of cavities in amorphous $SiO_2$ prepared by cold compression of SZ at 20 GPa is associated with the persistence of a large fraction of five-fold rings and that this is an important signature of permanent densification induced by both atomistic and topological order.

Here, we emphasise that permanent densification is only observed in amorphous $SiO_2$ obtained by the cold compression of bulk crystalline siliceous zeolite ($b$SZ). Because permanent densification was not achieved by cold compression of GS[19] or $p$SZ, it is expected that, in general, heating is important for permanent densification. The effects of temperature on compression of $SiO_2$ have been reported previous studies[18,19,37,38]. In addition to the compression of silica glass[18,19,37,38] relaxation of densified silica glass by thermal annealing has also been reported[38,39]. This expectation also raises the question of whether permanent densification persists even after annealing. To confirm the thermal stability of the densified samples, we heated cold-compressed $b$SZ to 750 °C. The effect of thermal annealing on $S(k)$ was apparent, as shown in Fig. 4a, b. As seen in Fig. 4b, the $S(k)$ of annealed amorphous $SiO_2$ is not identical to that of pristine GS below 6 Å$^{-1}$; there are differences in the FSDP heights and in the low-$k$ (small-angle) region below 1 Å$^{-1}$. Notably, the tiny sharp diffraction peaks in amorphous $SiO_2$ diminished after annealing at 750 °C, indicating that they were associated with SZ rather than impurities. Intriguingly, the GS densified by cold compression was converted into pristine GS using the same annealing process (Fig. S3). The recovered stishovite also became amorphous upon heating[40]. The change in $S(k)$ after annealing clearly demonstrates that permanent densification is maintained only at ambient temperatures, and that another metastable structure (topology) of densified amorphous $SiO_2$ is generated by thermal treatment. Such a transformation (relaxation) has also been observed in other papers[38,39], in which the saturation behaviour was dependent on the annealing temperature. Elucidating the key structural details necessary for maintaining a metastable densified $SiO_4$ network is the next goal for distinguishing between thermally metastable and reversible $SiO_4$ networks.

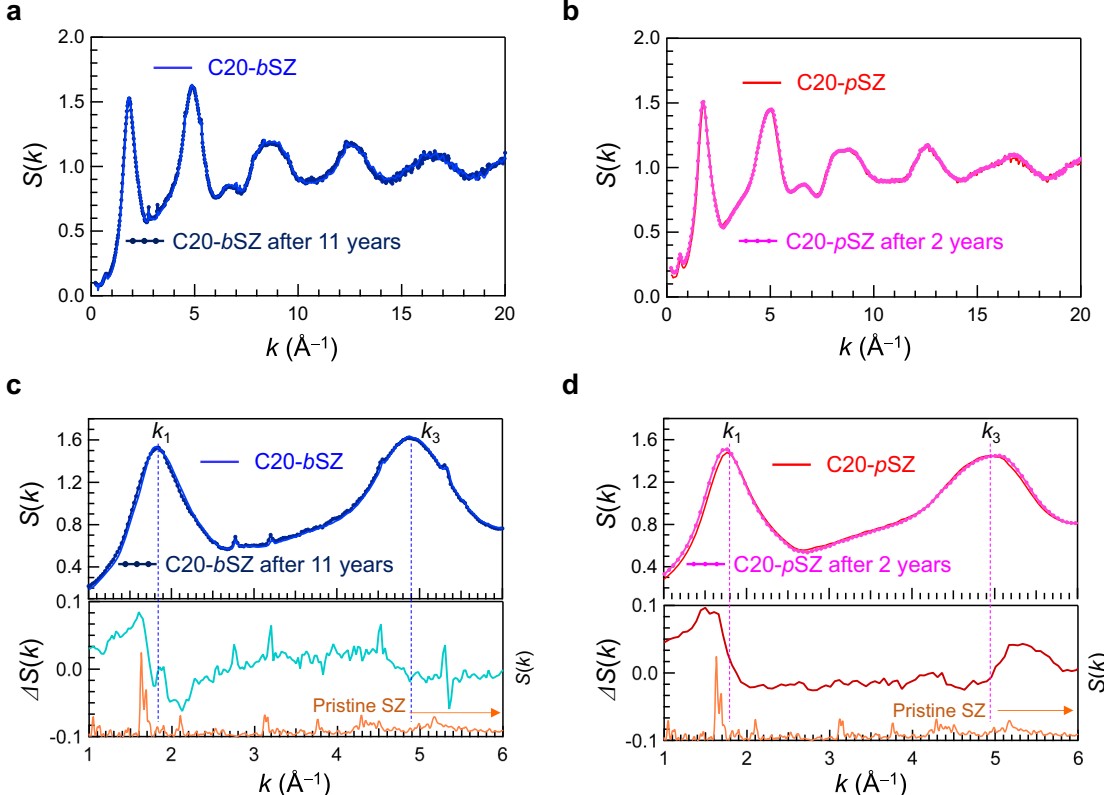

**Fig. 2 X-ray diffraction data showing the time-dependent variation of densified amorphous SiO$_2$. a** Total structure factors, $S(k)$, of densified amorphous SiO$_2$ prepared by 20 GPa-cold compressions obtained from bulk crystal siliceous zeolite (C20-$b$SZ). **b** Total structure factors, $S(k)$, of densified amorphous SiO$_2$ prepared by 20 GPa-cold compressions obtained from siliceous zeolite powder (C20-$p$SZ). **c, d** Enlarged $S(k)$ of amorphous SiO$_2$ and differential $S(k)$ between the as-prepared sample and the same sample after the stated elapsed time; the dashed lines indicate the positions of the $k_1$ (FSDP) and $k_3$.

**Effect of ball-milling on the structure of SZ.** Figure 5a shows the $S(k)$ values of the pristine SZ and ball-milled (BM) $p$SZ. The sharp Bragg peaks from the crystalline structure of the SZ disappeared after ball milling, indicating amorphization. In the diffraction pattern of the BM-SZ, the FSDP is at $k = 1.65 \text{ Å}^{-1}$, which is similar to the position of a peak observed for amorphous SiO$_2$ obtained by a compression at 7.7 GPa and RT. Also, a broad peak at $k = 5 \text{ Å}^{-1}$, which is conventionally attributed to $k_3$[22] in pristine GS, is observed. Notably, the scattering intensity increased remarkably compared to that of pristine GS in the low-$k$ region as a result of sample grinding. We also note that the Bragg peaks observed at $k \sim 0.6 \text{ Å}^{-1}$, which are associated with the zeolite cage in crystalline SZ, remarkably diminish in the BM-sample but do not completely disappear, suggesting that the cage breakdown is incomplete. Considering that the $k_0$ peak has the same $k$ value as the Bragg peaks, it is expected that the sample partially retains its crystallinity, although the cage structure appears to be much more disordered because of the lack of translational periodicity. Figure 5b, c shows the $^{29}$Si magic angle spinning (MAS) NMR spectra of the SZs before and after ball-milling, respectively. As indicated by dashed lines, each silicate Q$^n$ unit was separated by peak deconvolution. In the case of SZ before the treatment, the Q$^4$ peak observed at $-110$ ppm was asymmetric[41–44]. The chemical shift of the Q$^4$ peak in $^{29}$Si MAS NMR changes depending on the Si–O interatomic distance $r$ and $\rho$ ($=\cos\theta/(1-\cos\theta)$) determined by the Si–O–Si bond angle $\theta$[44,45]. Because the $G(r)$ of SZ exhibits a Si–O correlation represented by a single normal distribution similar to that of another zeolite[46], the asymmetry of the Q$^4$ peak in the MAS NMR spectrum should arise from the presence of sites in the SZ with different coupling angles. The fitting parameters for the materials used to analyse

the $G(r)$ and $^{29}$Si NMR spectra are listed in Tables S1 and S2, respectively. By calculating each Q$^n$ area, each Q$^n$ fraction is quantified by calculating its area. These data are presented in Table 2. Data for SiO$_2$ (Fig. S4), and SZ without BM treatment[47] were also included for comparison. Notably, the Q$^4$ peak at a higher magnetic field in the SZ disappears after ball milling. Considering the $S(k)$ and Q$^4$ peaks of the sample after ball-milling (Table S2), the Q$^4$ species at a higher magnetic field can be assigned to a silicate unit that contributes to the cage structure. Figure 5d shows the $S(k)$ values of BM-$p$SZ and cold-compressed amorphous SiO$_2$ from BM-$p$SZ by applying a pressure of 20 GPa and RT. $S(k)$ of pristine GS is also shown for comparison. Although FSDP and $k_3$ peaks were observed for amorphous SiO$_2$, both peak heights were lower than those of pristine GS. The $S(k)$ profiles of BM-SZ and the densified BM amorphous SiO$_2$ are similar to that of pristine GS at $k > 3 \text{ Å}^{-1}$, suggesting that the short-range structure of BM-SZ is also similar to that of amorphous SiO$_2$. Notably, a significant difference is observed in the low-$k$ region. Although the height of the small-angle scattering peak below $k \sim 1 \text{ Å}^{-1}$ in the milling-induced amorphous SZ sample decreases after densification, it does not completely disappear. Figure 5e shows $G(r)$ for all samples. The densities of ball-milled samples were estimated from the slope of the dotted lines using $\rho = \frac{1}{4\pi}\frac{\partial G(r)}{\partial r}$ (see Fig. S5 for details). We found that the density of the ball-milled amorphized SZ (2.2 g cm$^{-3}$) is higher than that of pristine SZ (1.8 g cm$^{-3}$) and comparable to that of GS (2.2 g cm$^{-3}$). After cold compression at 20 GPa, the density of the ball-milled amorphized SZ increases (2.4 g cm$^{-3}$). Based on these density values, we suggest that densification occurs through the collapse of zeolite pores following the breaking of the SZ cage during ball milling.

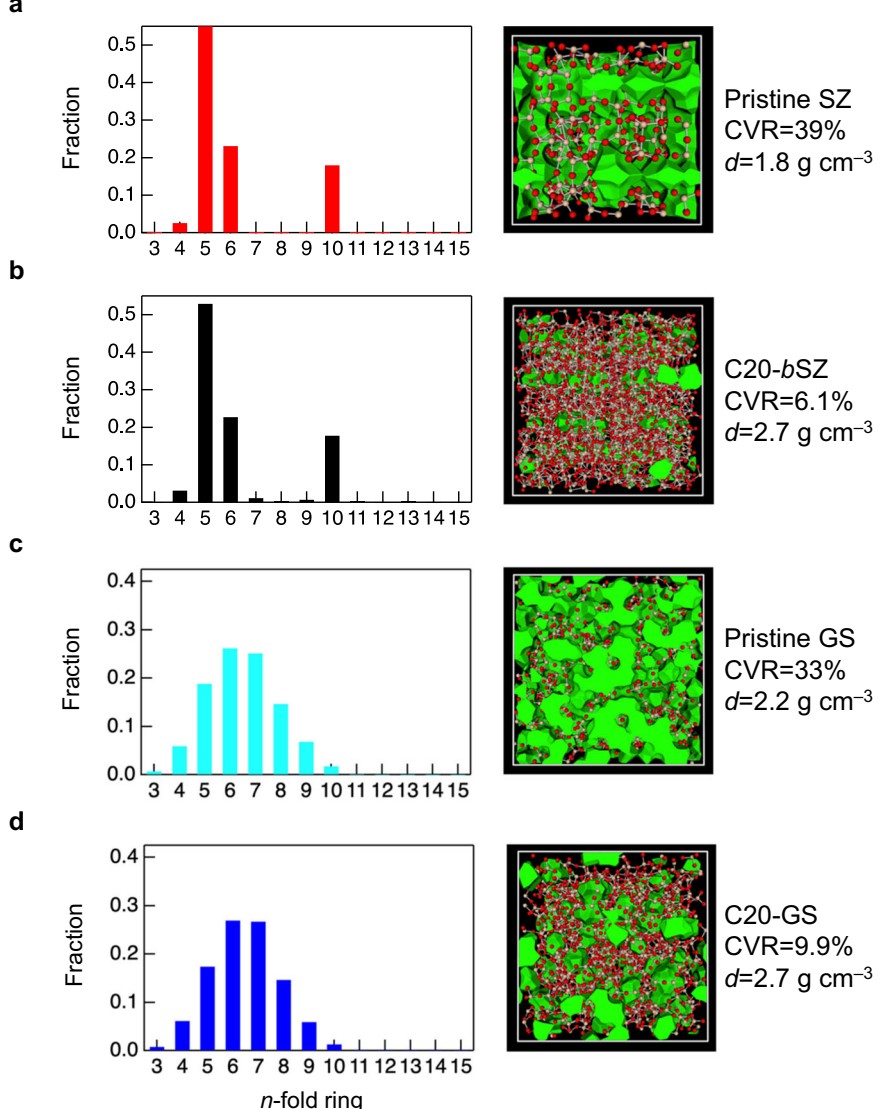

**Fig. 3 Topology of amorphous SiO$_2$ derived from siliceous zeolite (SZ) and glassy SiO$_2$ (GS). a** $n$-fold ring distribution of pristine SZ, **b** densified amorphous SiO$_2$ after cold compression with 20 GPa (C20-$b$SZ), **c** pristine GS, and **d** densified GS after cold compression with 20 GPa (C20-GS). The images on the right show the cavities in these SiO$_2$-based materials, indicating the densities and CVR ratios (green: cavity; orange: silicon; and red: oxygen).

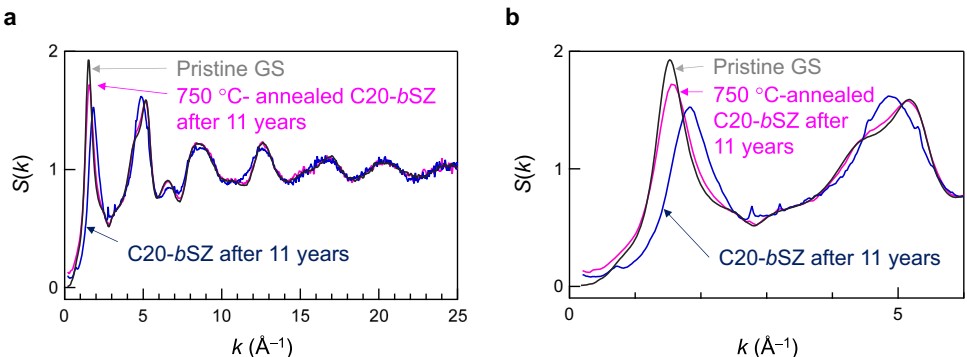

**Fig. 4 Effect of thermal annealing on structure of amorphous SiO$_2$ derived from siliceous zeolite (SZ). a** $S(k)$ of densified amorphous SiO$_2$ from $b$-SZ after cold-compressed with 20 GPa (C20-$b$SZ) and that of cold-compressed amorphous SiO$_2$ after thermal annealing at 750 °C for 1 h; the $S(k)$ of pristine GS is also shown for comparison. **b** Enlarged $S(k)$ of densified SZs plotted together with that of pristine GS.

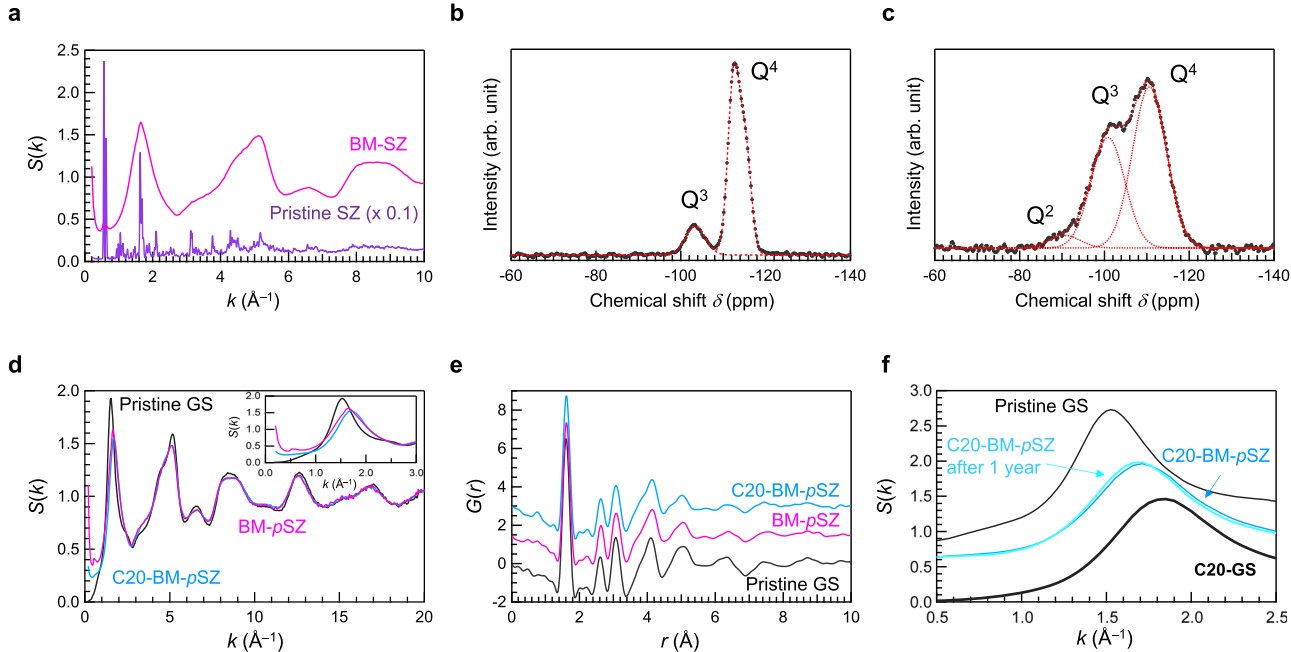

**Fig. 5 Effect of ball milling (BM) on the structure of siliceous zeolite (SZ). a** X-ray total structure factors, $S(k)$, of SZ with and without BM treatment. $^{29}$Si MAS NMR spectra of SZ (**b**) and SZ after BM-treatment (**c**). **d** X-ray total structure factors, $S(k)$, of BM-SZ and densified BM amorphous $SiO_2$ obtained by cold compression, shown together with that of pristine GS; inset: enlarged $S(k)$ at the FSDP region. **e** Reduced pair distribution functions, $G(r)$, of all samples shown in (**d**). **f** $S(k)$ of amorphous $SiO_2$ (BM) measured soon after cold pressing and again after 1 year along with $SiO_2$ and densified GS; successive BM-SZ and GS data are displayed upward at $S(k)$ for clarity.

**Table 2 Ratio of $Q^n$ units in siliceous zeolite (SZ) and glassy $SiO_2$ (GS) before and after ball-milling (BM).**

| Chemicals | Treatment | $Q^2$ | $Q^3$ | $Q^4$ |
|---|---|---|---|---|
| SZ | Before BM | 0 | 0.15 (±0.01) | 0.85 (±0.01) |
| | After BM | 0.04 (±0.01) | 0.39 (±0.01) | 0.57 (±0.01) |
| GS | Before BM[47] | 0 | 0 | 1.00 |
| | After BM | 0.06 (±0.01) | 0.53 (±0.02) | 0.41 (±0.02) |

**Comparison of amorphous $SiO_2$ in $k$ space.** Finally, to compare the pristine GS with other densified amorphous silicas prepared from SZs, the small-$k$ region of $S(k)$ of the BM-$p$SZ is depicted in Fig. 6, together with the data from previously reported $SiO_2$ materials. Vertical dashed lines A and B–D serve as visual guides to clarify the positions of $k_0$ and $k_1$, respectively. For amorphous $SiO_2$ derived from SZ, a peak at $k_0$, which is characteristic of SZ, was observed. Although the FSDP position is sensitive to pressure[48], a distinct correlation between the position of the FSDP and density was observed only in the GS and not in the SZ. However, if we exclude the BM-$p$SZs, we believe that a correlation exists between the density and FSDP position of amorphous $SiO_2$ prepared from SZs. Notably, the $k_0$ value of BM-$p$SZ, whose FSDP position was at the lowest wavevector $k$, was the lowest $k_0$ peak among these materials. In SZ-derived amorphous $SiO_2$, it is suggested that the values of the wave vectors $k_0$ and $k_1$ are correlated.

The obtained results show that the crystalline topology affects pressure-induced material fabrication and thermal stability. The amorphization of the SZ by cold compression is linked to the collapse of the pores in the SZ, and a trace of the cage structure has been already reported in a previous paper[31]. Notably, traces of the SZ remained in the densified amorphous $SiO_2$ after thermal annealing. These traces of SZ influence permanent densification and provide evidence of the structural differences between the

starting materials SZ and GS in amorphous $SiO_2$. Materials with the same chemical composition but different topologies can be fabricated by tailoring the starting materials, which will pave the way for the design of novel functional materials.

Amorphous materials prepared by applying high pressure to the SZ at room temperature using various treatments were characterised. The results confirm that the structural changes depend on the stabilisation treatment and pressurisation conditions. The X-ray structure factor $S(k)$ of the amorphous $SiO_2$ derived from a single crystal of SZ changed slightly over an 11-year period. In particular, it was found for the first time that samples prepared from SZs by high-pressure synthesis have a characteristic $k_0$ peak at a lower $k$ than that of FSDP, which is a remnant of some Bragg peaks of SZs. Furthermore, the $k_0$ peak is a disrupted structural motif of SZs or a long-distance correlation rather than a remnant Bragg peak, as the peak is shifted to a higher $k$. The $k_0$ peak, which is characteristic of a cage within the SZ, disappeared after mechanical ball milling. The results demonstrated that the topology of the pressure-induced amorphous materials could be tuned by tailoring the nature of the starting materials. We are confident that the clarification of the unique structure existing at distances beyond the intermediate will provide a guide for opening up a new science of amorphous materials.

## Materials and methods
**Sample details.** SZ powder was purchased from Tosoh Corp. (890HOA, MFI-type zeolite), and 890HOA was selected because its Si/Al ratio is sufficiently high (>1000) and its Al content is sufficiently low. The linear dimensions of the crystallites were 2–5 μm, and the material contained additional $H^+$ cations.

**Densification of samples.** The densified $SiO_2$ samples using SZ as a starting material were prepared using a Kawai-type apparatus with a Walker-module (mavo press LPR 1000-400/50; Max

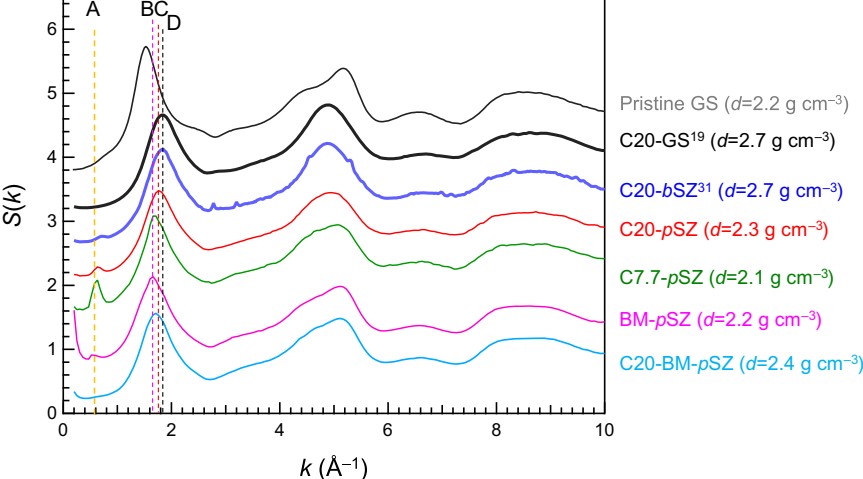

**Fig. 6 Comparison of $S(k)$ of densified GS derived from glassy $SiO_2$ (GS) and siliceous zeolite (SZ).** Enlarged $S(k)$ of densified GS and SZ with different pressures. $S(k)$ of pristine GS, 20 GPa cold compressed densified GS (C20-GS) and SZ (C20-SZ), ball-milled (BM)-SZ, and amorphous $SiO_2$ derived from ball-milled SZ by cold compression with 20 GPa (C20-BM-$p$SZ). The symbols A, B, C, and D indicate the $k$ values at the peak of $k_0$, the FSDP of ball-milled SZ, amorphous $SiO_2$ derived from SZ with an applied pressure of 20 GPa at RT, and densified GS after cold compression with 20 GPa, respectively.

Voggenreiter GmbH, Mainleus, Germany) at the Frontier Materials Laboratory, Tokyo Institute of Technology. The powdered samples that formed into pellets were sealed in a gold capsule and pressed at RT at an applied pressure of 20 GPa for 1 h. A 1500-ton belt-type high-temperature, high-pressure apparatus installed at the National Institute for Materials Science (NIMS), with an applied pressure of 7.7 GPa was used to prepare the samples. The strategy was as follows: (1) the powdered sample was moulded into a cylindrical shape with a diameter of 4 mm and height of 3 mm and (2) pressurised to 7.7 GPa in 5 h. Subsequently, (3) the applied pressure was maintained for 30 min, after which (4) the applied pressure was reduced to 0 GPa in 5 h. The application of high pressure at an ambient temperature is known as cold compression.

**Ball-milling treatments.** To obtain a less-ordered SZ, it was ground (by ball milling) in air at 500 rpm using a Fritsch P6 planetary ball-mill system, a silicon nitride pot, and silicon nitride balls. To prevent the pot from heating, the system was allowed to run for 15 min, and then stopped running for another 15 min to cool. Overall, grinding was performed for 24 h.

**Thermal annealing treatments.** To verify the permanent densification of C20-$b$SZ, the sample was heat treated in air in a commercially available electric furnace. The heating strategy was as follows: (1) the sample was heated to 750 °C at a heating rate of 10 °C/min, (2) 750 °C was maintained for 1 h for thermal annealing, following which (3) the sample was cooled to room temperature without the use of cooling-rate control.

**NMR measurements.** The local structures of the Si atoms in the pristine and ball-milled SZ were evaluated using $^{29}$Si MAS NMR spectroscopy (JEOL ECA 300 (7.1 T) spectrometer) at a Larmor frequency of 59.7 MHz. The sample powder was packed in a 4.0-mm $ZrO_2$ rotor and spun at 7.5 kHz. Single-pulse experiments were conducted using 30° pulses with a repetition delay of 20 s. Tetramethylsilane (TMS) was used as the reference material (0 ppm) to calibrate the $^{29}$Si chemical shift. To estimate the population and NMR parameters of each Si species, the spectra were fitted to Gaussian functions.

**High-energy XRD measurements.** High-energy XRD measurements were performed on the BL04B2 beamline at SPring-8

(Hyogo, Japan) using a two-axis diffractometer dedicated to studying disordered materials. The energy of incident X-rays was 61.34 keV. The raw data were corrected for polarisation, absorption, and background, and the contribution of Compton scattering was subtracted using a standard data analysis software. The corrected X-ray diffraction data were normalised to obtain the total structure factor, $S(k)$.

**Topological analyses.** Ring size distribution calculations were performed for SZ (reverse Monte Carlo (RMC) model)[49] and GS (molecular dynamics–RMC model)[19,22,50] using the R.I.N.G.S. code[51,52]. Cavity volume analysis was performed using PyMol-Dyn code[53]. The code can calculate three types of cavities: domain, centre-based (Voronoi), and surface-based cavities. We calculated the surface cavity volumes using a cutoff distance $r_c = 2.5$ Å.

## Data availability

All relevant data supporting the findings of this study are available from the corresponding author upon request.

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

## Acknowledgements

This work was partially supported by TIA Kakehashi TK19-004 (to S.K., Y.S., H.M., T.W., and S.S.). Support was also received from JSPS Grant-in-Aid for Transformative Research Areas (A) "Hyper-Ordered Structures Science" (grant numbers 20H05878, 20H05880, 20H05881 and 20H05882) and the Collaborative Research Projects of Laboratory for Materials and Structures, Institute of Innovative Research, Tokyo Institute of Technology. High-energy XRD measurements were performed at BL04B2 of SPring-8 with the approval of the Japan Synchrotron Radiation Research Institute (JASRI) (proposal numbers 2019B1563 and 2021A1166). We are grateful to F. Kawamura, M. Miyakawa, and T. Taniguchi (High-Pressure Structural Controls Group, NIMS) for their support with the high-pressure and high-temperature syntheses using a 1500-ton belt-type apparatus at NIMS.

## Author contributions

Design: H.M. and S.K.; Methodology: H.M. and S.K.; Investigation: H.M., S.K., Y.S., Y.O., A.M., S.S., K.O. and D.A.K.; Sample preparation: T.W., Y.Sh., Y.Sa., J.H., C.L., P.H., A.I. and M.A. Writing—original draft: H.M. and S.K.; Writing—review and editing: H.M., S.K., J.H. and D.A.K. All authors have read and agreed to the published version of the paper.

## Competing interests

The authors declare no competing interests.
