## [Peer Review File · Communications Chemistry]

Reviewers' comments:

Reviewer #1 (Remarks to the Author):

The paper "Siliceous zeolite-derived topology of amorphous SiO₂" reports interesting work and results on the densification of amorphous silica after cold compression comparing with siliceous zeolite. The manuscript could be considered for publication in the Journal after some revisions have been made as follows:

Concerning the results in Figure 3

- 1) Where can be found details about calculations ?
- 2) Used labels can lead to confusion i.e densified amorphous SiO₂ 20 GPa is not clear that obtained from zeolite. Please clarify the names given to samples.
- 3) It is a bit surprising that ring statistic after 20 GPa of cold compression remains constant. The authors need to discuss their results for instance with Guerette et al. 2015 Sc. Rep. where ring statistic is different as well as their evolution under 20 GPa, RT

P7: "it is expected that heating is important for permanent densification ", please explain this sentence. Do the author refer to hot compression. An abundant literature was published on this topic. From a general point of view, the discussion with other papers is quite poor, thermal annealing of cold compressed SiO₂ and comparison with hot compression was well discussed recently by many authors (Guerette, Cornet, ...).

Fig.2 why there is no overlapping of pristine / after 2 years/ after 11 years for more clarity. The S(k) shift value needs to be precised

Reviewer #2 (Remarks to the Author):

The authors present a very interesting paper on obtaining different forms of amorphous silica by using siliceous zeolite as the starting material. The paper demonstrates cold compression produces material with different topologies, and some are permanently densified while others are not. Overall, this is an important contribution to the field that should be published after considering the following points:

- 1) There are four main materials discussed in the paper: one produced by compression of a powdered siliceous zeolite; one produced by compression of a single crystal sample of siliceous zeolite; one prepared by ball milling siliceous zeolite; and finally, a pristine glassy silica sample for comparison. However, at times it is not always clear which of the zeolite derived samples is being referred to, and indeed whether what I have summarized here is correct. It would be useful if the authors gave each sample a clear recognizable name, i.e. aSZ1, and gave more details of how each sample was prepared in the "materials and methods" section. Examples of further information: how long the sample was ball-milled; which sample went to 7.7 GPa, and which went to 20 GPa; was just one loading used, or were multiple loadings/samples combined?

2) In figure 1, and the related text, a k_0 peak is introduced. The authors' point out this peak is in a similar position to the strongest peak of the crystalline starting materials, and then use it to argue the structure has therefore retained some of the zeolitic characteristics. How can they be sure the sample is completely amorphous, and that this is not indicating some very strain broadened crystalline material remains? This would be consistent with the peak being more prominent at lower pressures.

3) At times the language used could be more carefully chosen. For example, in the final sentence on page 5 it is stated "Although the two sets of data appear identical...", "very similar" or something would be more appropriate since the differences in the two data sets are then discussed. Again, on page 9 "The $S(k)$ profiles of BM siliceous zeolite and the densified BM amorphous SiO_2 are similar to that of glassy SiO_2 at $k > 3 \text{ \AA}^{-1}$, suggesting that the short-range structure of ball-milled siliceous zeolite is identical to that of amorphous SiO_2 ." Surely if the data are similar, the short-range structure would also be similar (not identical); or you could say consisting of the same structural units.

4) On Page 6 "Therefore, we can conclude that b-siliceous-zeolite-derived amorphous SiO_2 was permanently densified, whereas the amorphous SiO_2 from p-siliceous zeolite was not. Considering that densified glassy SiO_2 from bulk glassy SiO_2 exhibits permanent densification behavior with thermal treatment¹⁹, we assume that monolithic materials are important starting materials for sustaining permanent densification by cold compression." It is not clear what "monolithic materials" refers to here, do the authors' mean a single crystal sample is important for permanent densification, or does this refer to sample purity or other property?

5) On page 10 "Furthermore, the k_0 peak is a remnant structural motif or long distance correlation rather than a remnant Bragg peak as the peak is shifted to higher k ." The fact the peak has shifted does not rule out it being a remnant Bragg peak, the material is clearly under extreme strain following compression and this would reduce the crystalline cell parameter and so the peak would shift.

6) More details on how the analysis presented in Figure 3 was carried out would be useful; was modelling used to produce the images on the right?

7) Finally, was the data modelled with different amorphous structures to demonstrate that the zeolite derived material truly retains its cage-like structural motifs? This would make the overall argument of the paper much stronger.

Reviewer #3 (Remarks to the Author):

Review of manuscript COMMSCHEM-23-0383-T

Title: Siliceous zeolite-derived topology of amorphous SiO_2

The paper describes the influence of several factors on the topology, durability and degree of amorphization of SiO_2 . Duration of up to 11 years, impact of high pressure up to 20 GPa, ball milling and effect of annealing were verified.

The paper is short and concise, and after some additions regarding experimental work, fits in my opinion with the editorial policy of a journal like Communications Chemistry. I have reported several comments throughout the manuscript, embedded in the pdf file, that I hope will be useful in order to improve this study.

I also have two substantive comments:

It is not clear how the topology (Fig 3) of studied material was determined. Previous studies report

molecular dynamics and fitting to the XRD patterns. Here it was not stated neither in the text or experimental section. It should be clarified

I find it very interesting to use NMR in this study. Unfortunately, as a non-expert in this field I was not satisfied with the provided information. The spectrum changed compared to starting material. The provided reference no. 38 did not help in the understanding of the peak assignment. I suggest adding some background information on the supplementary material regarding this method and results.

Some additional general comments:

I suggest adding ordering numbers in Table 1, and use them as abbreviations for the materials names to save space in the main text.

I am not a native speaker and felt quite a few sentences being too long, exceeding 40 words. It is much easier to follow and keep flow of reading when sentences are shorter 15-20 words. I suggest to reword some of the text accordingly. For example:

“In the case of the amorphous SiO₂ obtained from b-siliceous zeolite (Fig. 2C), the difference spectrum around these two peaks is symmetric (i.e., no peak shift), clearly indicating that the S(k) profile of amorphous SiO₂ derived from b-siliceous-zeolite remained the same after 11 years.”

Could be changed to shorter, for instance:

The amorphous SiO₂ from b-siliceous zeolite is very stable, shows no difference in the diffraction pattern after 11 years (i.e., no peak shift Fig. 2C).

Response letter of COMMSCHEM-23-0383A

We really appreciate valuable comments from the reviewer #1, #2 and #3.

We totally revised the manuscript with following the reviewer's comments. The revised parts are highlighted with red colour. All figures and Tables are revised using ID of each sample.

To Reviewer #1,

The paper "Siliceous zeolite-derived topology of amorphous SiO₂" reports interesting work and results on the densification of amorphous silica after cold compression comparing with siliceous zeolite.

The manuscript could be considered for publication in the Journal after some revisions have been made as follows:

Concerning the results in Figure 3

1) Where can be found details about calculations ?

We have added the description in the "materials and methods" section.

Topological analyses. The ring size distribution calculations were performed for SZ (reverse Monte Carlo (RMC) model)⁴⁵ and glassy GS (molecular dynamics–RMC model)^{9,22,45} by the R.I.N.G.S. code^{46,47}. A cavity volume analysis was performed using the pyMolDyn code⁴⁸. The code can calculate three different types of cavities, domain, center-based (voronoi), and surface-based cavities. We calculated surface cavity volumes with a cut off distance $r_c=2.5$ Å.

2) Used labels can lead to confusion i.e densified amorphous SiO₂ 20 GPa is not clear that obtained from zeolite. Please clarify the names given to samples.

We appreciate suggestion by the reviewer #1. The names listed in Table 1 have been revised.

3) It is a bit surprising that ring statistic after 20 GPa of cold compression remains constant. The authors need to discuss their results for instance with Guerette et al. 2015 Sc. Rep. where ring statistic is different as well as their evolution under 20 GPa, RT

We appreciate your good suggestion. As the reviewer suggested, the results of GS seem to be conflicted with previous results on densified SiO₂ glass (ref.37). However,

although the definition of an n -membered ring and the simulation method of the present study are different from previous one, we assume that it is difficult to conclude that remarkable difference of ring structure is observed by cold compression of GS.

P7: "it is expected that heating is important for permanent densification ", please explain this sentence. Do the author refer to hot compression. An abundant literature was published on this topic.

We appreciate reviewer's important suggestion. We have added other previous papers on SiO₂ by hot compression.

From a general point of view, the discussion with other papers is quite poor, thermal annealing of cold compressed SiO₂ and comparison with hot compression was well discussed recently by many authors (Guerette, Cornet, ...).

We appreciate your important suggestion. We have added other previous papers on cold & hot compression of SiO₂. Papers on annealing of densified SiO₂ glass are also added in reference.

Fig.2 why there is no overlapping of pristine / after 2 years/ after 11 years for more clarity. The S(k) shift value needs to be precised

We appreciate the good suggestion. We have added XRD of pristine siliceous zeolite in both Figs. 2c and 2d.

To Reviewer #2,

The authors present a very interesting paper on obtaining different forms of amorphous silica by using siliceous zeolite as the starting material. The paper demonstrates cold compression produces material with different topologies, and some are permanently densified while others are not. Overall, this is an important contribution to the field that should be published after considering the following points:

There are four main materials discussed in the paper: one produced by compression of a powdered

siliceous zeolite; one produced by compression of a single crystal sample of siliceous zeolite; one prepared by ball milling siliceous zeolite; and finally, a pristine glassy silica sample for comparison. However, at times it is not always clear which of the zeolite derived samples is being referred to, and indeed whether what I have summarized here is correct. It would be useful if the authors gave each sample a clear a recognizable name, i.e. aSZ1, and gave more details of how each sample was prepared in the “materials and methods” section. Examples of further information: how long the sample was ball-milled; which sample went to 7.7 GPa, and which went to 20 GPa; was just one loading used, or were multiple loadings/samples combined?

We appreciate suggestion by the reviewer #2. The ID of each sample has added. In addition, the details of cold compression, ball milling, and RMC modelling have been written in the “materials and methods” section. The added sentences are shown in the red colour in the revised manuscript.

2) In figure 1, and the related text, a k_0 peak is introduced. The authors' point out this peak is in a similar position to the strongest peak of the crystalline starting materials, and then use it to argue the structure has therefore retained some of the zeolitic characteristics. How can they be sure the sample is completely amorphous, and that this is not indicating some very strain broadened crystalline material remains? This would be consistent with the peak being more prominent at lower pressures.

We can be sure that the sample is completely amorphous as the k_0 peak disappeared upon heating at 750 °C. It is well known that the crystalline zeolite is stable up to 1200 °C, at which temperature it transforms to cristobalite. If the peak came from remaining crystalline material, the peaks should sharpen on heating due to relaxation of any strain present.

3) At times the language used could be more carefully chosen. For example, in the final sentence on page 5 it is stated “Although the two sets of data appear identical...”, “very similar” or something would be more appropriate since the differences in the two data sets are then discussed. Again, on page 9 “The $S(k)$ profiles of BM siliceous zeolite and the densified BM amorphous SiO_2 are similar to that of glassy SiO_2 at $k > 3 \text{ \AA}^{-1}$, suggesting that the short-range structure of ball-milled siliceous zeolite is identical to that of amorphous SiO_2 .” Surely if the data are similar, the short-range structure would also be similar (not identical); or you could say consisting of the same structural units.

We appreciate the comments from reviewer. The “identical” concerning to $S(k)$ has been

changed to similar.

4) On Page 6 “Therefore, we can conclude that b-siliceous-zeolite-derived amorphous SiO₂ was permanently densified, whereas the amorphous SiO₂ from p-siliceous zeolite was not. Considering that densified glassy SiO₂ from bulk glassy SiO₂ exhibits permanent densification behavior with thermal treatment¹⁹, we assume that monolithic materials are important starting materials for sustaining permanent densification by cold compression.” It is not clear what “monolithic materials” refers to here, do the authors’ mean a single crystal sample is important for permanent densification, or does this refer to sample purity or other property?

We think that single crystal sample is important for permanent densification.

5) On page 10 “Furthermore, the k_0 peak is a remnant structural motif or long distance correlation rather than a remnant Bragg peak as the peak is shifted to higher k .” The fact the peak has shifted does not rule out it being a remnant Bragg peak, the material is clearly under extreme strain following compression and this would reduce the crystalline cell parameter and so the peak would shift.

We appreciate the comments from reviewer. We agree the comment that cell parameter has been changed by compression. The sentence has been changed as follows.

k_0 peak is a disrupted structural motif of siliceous zeolites,

6) More details on how the analysis presented in Figure 3 was carried out would be useful; was modelling used to produce the images on the right?

We have added the description in the “materials and methods” section.

Topological analyses. The ring size distribution calculations were performed for siliceous zeolite⁴⁵ and glassy SiO₂^{19, 22, 45} by the R.I.N.G.S. code^{46,47}. A cavity volume analysis was performed using the pyMolDyn code⁴⁸. The code can calculate three different types of cavities, domain, center-based (voronoi), and surface-based cavities. We calculated surface cavity volumes with a cut off distance $r_c=2.5$ Å.

7) Finally, was the data modelled with different amorphous structures to demonstrate that the zeolite derived material truly retains its cage-like structural motifs? This would make the overall argument of the paper much stronger.

We have added the following description in the main text (p. 9).

The amorphization of SZ by cold compression is linked to the collapse of the pores in the SZ, and the trace of cage structure has been already reported in the previous paper (ref. 31).

To Reviewer #3,

The paper describes the influence of several factors on the topology, durability and degree of amorphization of SiO₂. Duration of up to 11 years, impact of high pressure up to 20 GPa, ball milling and effect of annealing were verified.

The paper is short and concise, and after some additions regarding experimental work, fits in my opinion with the editorial policy of a journal like Communications Chemistry. I have reported several comments throughout the manuscript, embedded in the pdf file, that I hope will be useful in order to improve this study.

Response to comments embedded in the PDF file.

We appreciate the valuable comments embedded in the PDF. These responses are shown below.

p. 4

powder diffraction pattern of siliceous zeolite,
structure factor,

We have changed sentences with “powder diffraction pattern of siliceous zeolite” and “structure factor” in accordance with the reviewer’s comments.

p. 5

explain the symbols

Add the explanations of symbols.

The position of k_0 is either similar or the same. Can't be both. I suggest to remove second sentence. In the first sentence change same to similar.

According to the reviewer’s comments, the sentences have been revised.

The figure caption indicates that the comparison is between two siliceous materials: p-siliceous zeolite and p-siliceous zeolite. There is no data on glassy material in Fig 2.

Since there is no data, the word “glassy” has been removed.

p. 6

k_3 is not defined anywhere so far for siliceous zeolite. Supplementary materials define k_3 as ca 15Å-1, however for glassy material.

Since the chemical composition of glassy SiO₂ and that of siliceous zeolite are identical (SiO₂), the peaks can be discussed with the same manner.

What is k_3 . Is k_3 equivalent to PP?

NO, k_3 is a broad peak located higher k value than k_2 . k_3 is not equivalent to PP. Several sentences have been corrected.

There is no comparison of glassy and siliceous SiO₂ in Fig2. Such overlay is shown on Fig 4, however for thermal annealing influence.

The comparison of glassy and siliceous SiO₂ is shown in Fig. 1. The sentence has been revised.

It is not adequately explained how the ring size distribution was calculated. How the structural information was obtained? For example, Molecular Dynamics calculations? There is no experimental section regarding this kind of calculations.

We have added the description in the “materials and methods” section.

Topological analyses. The ring size distribution calculations were performed for SZ (reverse Monte Carlo (RMC) model)⁴⁵ and glassy GS (molecular dynamics–RMC model)^{9,22,45} by the R.I.N.G.S. code^{46,47}. A cavity volume analysis was performed using the pyMolDyn code⁴⁸. The code can calculate three different types of cavities, domain, center-based (voronoi), and surface-based cavities. We calculated surface cavity volumes with a cut off distance $r_c=2.5$ Å.

Was it ambient pressure(10⁻⁴ GPa) or was it really in vacuum?

The applied pressure was changed to 0 GPa. Thus, the sentence has been revised.

P. 7

Do you refer to previous studies? Add reference. In the manuscript there is no data presented that indicates this conclusion.

The reference (Ref. 19) has been added.

P. 8

Consider putting this paragraph to the Introduction

The sentences have been moved to introduction.

Figures are defined by small letters a,b,c... In the text capital letters are used Fig. 5A. The captions should be unified throughout the text.

We have corrected these symbols.

The intensity is remarkably increased, but relative to what

The intensity of ball-milled samples increased compared to that of pristine glassy SiO₂.

P. 9

Either mark them on the Figure 5d or give values of k (A-1) in brackets for FSDP and k3, that you refer to.

correct formatting

We have corrected font size.

define which exactly with a value of k in A-1

The apparent wavevector region has been added.

Formatting of the figure should be corrected

We have corrected colour of text.

P. 10

Change to Capital letters

We have corrected these symbols.

An information that k_1 means the same as FSDP should appear at the beginning of the text.

We have already mentioned as follows in p. 5. The sentence has been modified.

This peak can be referred to as k_0 because the FSDP on the higher- k side is referred to as k_1 and the second principal peak as k_2 ³².

change to "believe"

The word has been changed.

the call of the sentence implies that k_0 is the lowest k_1 . The sentence should be corrected

The k_0 value of the BM siliceous zeolite is the lowest wavevector among these materials exhibiting the k_0 peak. The sentence has been corrected.

I do not feel convinced that k_0 value is related to density. The data summarised in Table 1 do not prove such a relationship.

We agree that the relationship between k_0 and density is not clarified. The description has been removed.

New paragraph

New paragraph has been made.

11?

We have corrected as 11.

P. 12

this sample designation does not appear anywhere in the text

We have corrected name of material.

Change to $S(k)$

We have corrected as $S(k)$.

Response two comments (shown in the decision letter):

It is not clear how the topology (Fig 3) of studied material was determined. Previous studies report molecular dynamics and fitting to the XRD patterns. Here it was not stated neither in the text or experimental section. It should be clarified.

We have added the sentences in the “materials and methods” section.

Topological analyses. The ring size distribution calculations were performed for SZ (reverse Monte Carlo (RMC) model)⁴⁵ and glassy GS (molecular dynamics–RMC model)^{9,22,45} by the R.I.N.G.S. code^{46,47}. A cavity volume analysis was performed using the pyMolDyn code⁴⁸. The code can calculate three different types of cavities, domain, center-based (voronoi), and surface-based cavities. We calculated surface cavity volumes with a cut off distance $r_c=2.5$ Å.

I find it very interesting to use NMR in this study. Unfortunately, as a non-expert in this field I was not satisfied with the provided information. The spectrum changed compared to starting material. The provided reference no. 38 did not help in the understanding of the peak assignment. I suggest adding some background information on the supplementary material regarding this method and results.

We have added the reference (Ref. 46). The peak assignment of NMR is added in the supplementary material in addition to a supplementary reference.

Some additional general comments:

I suggest adding ordering numbers in Table 1, and use them as abbreviations for the materials names to save space in the main text.

We appreciate the reviewer's comments. The IDs have been used.

I am not a native speaker and felt quite a few sentences being too long, exceeding 40 words. It is much easier to follow and keep flow of reading when sentences are shorter 15-20 words. I suggest to reword some of the text accordingly. For example:

“In the case of the amorphous SiO₂ obtained from b-siliceous zeolite (Fig. 2C), the difference spectrum around these two peaks is symmetric (i.e., no peak shift), clearly indicating that the S(k) profile of amorphous SiO₂ derived from b-siliceous-zeolite remained the same after 11 years.”

Could be changed to shorter, for instance:

The amorphous SiO₂ from b-siliceous zeolite is very stable, shows no difference in the diffraction pattern after 11 years (i.e., no peak shift Fig. 2C).

We appreciate the important comments from the reviewer #3. The authors have shortened sentences as possible. The names listed in Table 1 have been revised for clarity.

REVIEWERS' COMMENTS:

Reviewer #1 (Remarks to the Author):

The authors follow the recommendations of the reviewers correctly and the article was enough improved to be published.

Reviewer #2 (Remarks to the Author):

The authors have addressed all the points raised and the manuscript should be published in its new form.

Reviewer #3 (Remarks to the Author):

All the major points have been addressed in the revised manuscript. I enclose some additional minor comments embedded in the pdf file.
However, the language could still be improved. You may want to consider professional editing services.

Thermal annealing description is missing including the way of cooling/quenching.

Response letter of COMMSCHEM-23-0383B

We really appreciate positive comments from the reviewer #1, #2 and #3.
According to the reviewer #3, the manuscript has been checked by the professional English Editing service. The certificate is attached at the end of this response letter.

To Reviewer #3,

All the major points have been addressed in the revised manuscript.

We appreciate positive evaluation by the reviewer #3.

I enclose some additional minor comments embedded in the pdf file.

Response to comments embedded in the PDF file.

We appreciate the valuable comments embedded in the PDF. These responses are shown below.

p. 5

define abbreviation

According to the reviewer's comments, the abbreviation has been added.
reduced pair distribution functions $G(r)$

p. 6

no

According to the reviewer's comments, the sentences have been revised.

This name brings confusion, what monolythic materials are. I suggest to change to single crystals as explained in the Response letter

According to the reviewer's comments, the sentence has been revised as follows.
we assume that single crystals are an important starting point for sustaining permanent densification via cold compression.

Compression

We have corrected the sentence.

p. 7

by 3.8%

The suggested words have been added.

(Removed) was

We have corrected the sentence in accordance with the reviewer's comments.

P. 8

G(r) appears earlier in the text

As shown above, the definition is shown in earlier in the text.

P. 11

thermal annealing description is missing and how fast the samples were cooled down/quenched?

We have added the description in the "materials and methods" section.

Thermal annealing treatments. To verify the permanent densification of C20-bSZ, the sample was heat treated in air in a commercially available electric furnace. The heating strategy was as follows: (i) the sample was heated to 750 °C at a heating rate of 10 °C/min, (ii) 750 °C was maintained for 1 h for thermal annealing, following which (iii) the sample was cooled to room temperature without the use of cooling-rate control.

However, the language could still be improved. You may want to consider professional editing services.

We would like to thank reviewer #3 for her/his important suggestions to improve the quality of the paper. The Editing certificate is attached below.

editage | helping you get published

Since 2002, Editage has helped over 430,000 authors publish around 1.2 million research papers in scholarly journals across over 1000 disciplines through editorial, translation, transcription, and publication support services. Editage is a brand of Cactus Communications (cactusglobal.com), a science communication and technology company.

GLOBAL : +1(833) 979-0061 | request@editage.com | **JAPAN :** 0120-00-2987 | submissions@editage.com

CACTUS

editage.com | editage.co.kr | editage.jp | editage.cn | editage.com.br | editage.com.tw | editage.de

Thermal annealing description is missing including the way of cooling/quenching.

We have added the description in the “materials and methods” section.

Thermal annealing treatments. To verify the permanent densification of C20-bSZ, the sample was heat treated in air in a commercially available electric furnace. The heating strategy was as follows: (i) the sample was heated to 750 °C at a heating rate of 10 °C/min, (ii) 750 °C was maintained for 1 h for thermal annealing, following which (iii) the sample was cooled to room temperature without the use of cooling-rate control.